# The Impact of an Intensive Care Diary on the Psychological Well-Being of Patients and Their Family Members: Longitudinal Study Protocol

**DOI:** 10.3390/healthcare11182583

**Published:** 2023-09-19

**Authors:** Vincenzo Bosco, Annamaria Froio, Caterina Mercuri, Vincenza Sansone, Eugenio Garofalo, Andrea Bruni, Assunta Guillari, Daniela Bruno, Michaela Talarico, Helenia Mastrangelo, Federico Longhini, Patrizia Doldo, Silvio Simeone

**Affiliations:** 1Department of Medical and Surgical Sciences, University Hospital Mater Domini, Magna Graecia University, 88100 Catanzaro, Italy; vincenzo.bosco@unicz.it (V.B.); froioannamaria@libero.it (A.F.); eugenio.garofalo@unicz.it (E.G.); andreabruni@unicz.it (A.B.); flonghini@unicz.it (F.L.); 2School of Medicine and Surgery, Magna Graecia University, 88100 Catanzaro, Italy; c.mercuri@unicz.it (C.M.); michaela.talarico@studenti.unicz.it (M.T.); 3Department of Experimental Medicine, Luigi Vanvitelli University, 80131 Naples, Italy; vincenza.sansone@unicampania.it; 4Department of Public Health, University of Naples Federico II, 80138 Naples, Italy; 5ASP Catanzaro, 88100 Catanzaro, Italy; helenia.mastrangelo@studenti.unicz.it; 6Clinical and Experimental Medicine Department, Magna Graecia University, 88100 Catanzaro, Italy; doldo@unicz.it (P.D.); silvio.simeone@unicz.it (S.S.)

**Keywords:** ICU, diary, post-traumatic stress disorder, anxiety, depression, quality of life

## Abstract

Background: Thanks to medical and technological advancements, an increasing number of individuals survive admission to intensive care units. However, survivors often experience negative outcomes, including physical impairments and alterations in mental health. Anxiety, depression, cognitive impairments, post-traumatic stress disorders, and functional disorders are known collectively as post-intensive care syndrome (PICS). Among the key triggering factors of this syndrome, memory impairment appears to play a significant role. Aims: This study aims to evaluate the impact of an intensive care diary on the psychological well-being of patients and their relatives after discharge from the ICU. Design: Prospective observational study. Expected results: The results of this study evaluate the impact of an ICU diary on the quality of life of ICU survivors and their family members.

## 1. Background

Thanks to technological advancements and evolving medical knowledge, an increasing number of individuals requiring admission to intensive care units are surviving [1,2]. Approximately 50–70% of survivors experience long-term cognitive, psychological, and functional impairments as a result of their ICU admission [3]. After ICU discharge, survivors often face adverse outcomes in physical function, cognitive function, and mental health status [4].

Physical consequences, such as limitations in physical function, are common and can significantly impact a patient’s life and level of independence [5,6]. Psychosocial issues are frequently observed and can range from mild to severe symptoms [6]. These disorders are collectively known as post-intensive care syndrome (PICS) [7,8]. Furthermore, 50% of patients who survive ICU admission do not return to school or work until one year later [9], and many do not regain their initial health status [10].

Psychological disorders related to PICS, including Post-Traumatic Stress Disorder (PTSD), depression, and anxiety, can occur in up to a fifth of survivors during the first 12 months after ICU discharge and are associated with a poorer quality of life [11,12,13]. Additionally, up to one-third of ICU survivors report anxiety [14] and depression [15]. PICS-related disorders can manifest even up to 15 years after surviving an ICU admission [16]. Memory impairment appears to play a significant role in the development of PTSD following ICU admission [17,18]. 

The absence of memories related to the ICU stay is associated with more severe PTSD symptoms [19]. Furthermore, patients with prolonged ICU stays often experience not only memory gaps but also delirium and hallucinations [20]. These delirious memories of the ICU experience are associated with the development of PTSD in survivors [3]. Post-discharge sequelae have a substantial impact on the financial situation of patients and their partners [21]. Family members of patients admitted to the ICU also report PTSD symptoms related to their loved one’s experience [22,23]. When a close family member is affected by critical illness, relatives often experience worry about the patient’s survival [24]. Family members subsequently become caregivers, and stressful events can further worsen their quality of life [25,26]. Patient diaries [27] can be helpful for both ICU survivors and their relatives [3]. A strategy to reduce occurrence of these symptoms is to provide a reconstruction of the events [28].

In particular, during the COVID-19 pandemic, due to restrictions on family visitation implemented to prevent disease transmission, it was observed that the absence of informal communication between COVID-19 patients and their relatives led to increased stress among the relatives [29]. To foster a positive connection among patients, their families, and healthcare professionals during the COVID-19 pandemic, the use of diaries has been recommended as one of the strategies to enhance communication with vulnerable relatives and mitigate the post-ICU burden [30]. 

Research has shown the significant value of ICU diaries as a tool for processing emotions, gaining insight, reducing stress, documenting essential information, and facilitating communication between staff and patients [31,32,33,34,35,36]. Diary writing has provided ICU staff with a valuable means to reflect on their daily work and the care of critically ill patients as unique individuals. This remains true even in the midst of the COVID-19 pandemic, characterized by resource constraints and stringent family visitation restrictions, regardless of whether the entries are later perused by patients or their families [34]. ICU diaries are typically written at the patient’s bedside and serve as a record of their ICU admission, including the events leading to admission, the patient’s condition, and daily procedures or treatments. It is important to use simple, everyday language rather than technical or medical terms [37]. The purpose of the diary is to provide a clear narrative of the non-clinical events that occurred during the ICU stay. The involvement of family members, healthcare personnel, and, if possible, the patients themselves in writing the diary helps ICU survivors fill gaps in their memory [24,25,26,27,37,38,39,40], come to terms with their illness, and “reduce the impact or prevalence of imaginary events and hallucinations” [41,42]. 

ICU diaries are used to help patients make sense of their ICU experience and fill in memory gaps [19]. Additionally, these diaries provide a sense of control for relatives, allowing them to keep track of general events and document their support for their loved one when direct communication is not possible [43]. 

Healthcare professionals also appear to benefit from these diaries, as they have a positive impact on communication with family members [44,45] and facilitate the delivery of personalized care [46]. 

Since the publication of what appears to be the first study about the use of diaries in intensive care [47] to date [48], the use of diaries has been implemented in Europe [49] and extended to other continents [50]. In settings where this tool is available, patients and families often request the diary from the beginning of the hospitalization period [38]. It is recommended that the diary be written collaboratively by all key stakeholders, including patients, family members, and healthcare personnel [51].

Despite the usefulness of diaries for patients, family members, and healthcare professionals [52,53] in preventing or reducing psychological problems, such as symptoms of PTSD, anxiety, and depression [3,9], the literature still presents different opinions regarding their effectiveness in improving recovery after critical illness [19,54]. In an intensive care setting, patients and families come from diverse backgrounds and have different critical illnesses [55]. Insufficient data exist to determine whether ICU diaries represent the most appropriate intervention to promote psychological health [56]. Additionally, to the best of our knowledge, only one qualitative study conducted in a specific Italian region has evaluated their use, focusing solely on diaries written by family members [57].

Iannuzzi’s research [28] stands as the only implementation study conducted within an Italian hospital to explore the utilization of diaries. 

For these reasons, as stated by the same authors, we believe it is essential to conduct a new study on the use of diaries in Italian intensive care unit.

## 2. Study Objective

### 2.1. The Aims of This Study Are as Follows

To assess the impact of an intensive care diary on the psychological well-being of patients and their relatives at 0, 3, and 12 months after discharge from the ICU.

The study has the following specific aims:

(1a) To test the hypothesis that an “intensive care diary” can reduce the patient’s risk of developing symptoms of PTSD, anxiety-related disorders, and depressive symptoms;

(1b) To test the hypothesis that an “intensive care diary” can reduce the risk of developing symptoms of PTSD, anxiety disorders, and depressive symptoms in the relatives of patients admitted to the ICU;

(2) To evaluate how the QoL trajectory of patients discharged from the ICU and their families may be influenced by the use of an “intensive care diary”;

(3) To explore the content and narrative structure of the diaries written by patients, relatives, and health professionals;

(4) To explore the experiential aspects related to use of a diary in intensive care.

### 2.2. Hypothesis

Patient diaries can provide assistance to ICU survivors and their families during their stay in the ICU amidst a critical illness. Additionally, they appear to offer benefits for healthcare personnel by fostering positive communication between healthcare providers and patients, thus facilitating the provision of personalized care.

Reconstructing memory of events that occurred during hospitalization in intensive care, through the use of a diary, can reduce onset of PTSD in subjects who survived ICU recovery.

The use of a critical care diary, in which daily events can be recorded by family members and caregivers, can assist in reducing the occurrence of PTSD, anxiety, depressive symptoms, and it can promote psychological well-being in both patients and their families.

Understanding how the use of diaries is experienced by participants can help develop educational programs aimed at implementing the use of this tool to improve their quality of life.

### 2.3. Design

The study will be conducted using a mixed methods approach [58], to explore the relationship between the use of diaries (qualitative) and the perceived QoL trajectory after ICU discharge (quantitative) [28]

Data collection is planned at 0, 3, and 12 months after ICU discharge. In the quantitative approach, descriptive observational methods will be employed using different types of questionnaires. In the qualitative approach, an interpretive narrative research technique will be used to analyze the diaries [59,60]. 

An interpretive approach will guide the data analysis process [61]. Furthermore, in order to explore the experience and the meaning attributed to the diaries, Cohen’s phenomenology [62] will also be used. This methodology combines Husserl’s descriptive and Gadamerian interpretative phenomenology. It was chosen as it has already been used, given its ability to facilitate a deeper understanding of both lived experiences and the meanings attributed to these experiences.

### 2.4. Inclusion Criteria

Patients:AGE > 18ICU length of stay > 72 hSpeaking and understanding the Italian languageConsent form

Family members:AGE > 18Family member ICU length of stay > 72 hSpeaking and understanding the Italian languageConsent form

Healthcare providers:AGE > 18Work in ICUSpeaking and understanding the Italian languageConsent form

### 2.5. Exclusion Criteria

For patients:Severe organ failurePre-existing presence of psychological pathology

For family members:Pre-existing presence of psychological pathology

#### 2.5.1. Setting and Data Collection Procedure

Data collection instruments will be administered at the time of discharge and then again at 3 and 12 months thereafter (Table 1).

#### 2.5.2. Data Collection

Patient

Socio-demographic questionnaire.Diary: A small lined notebook placed at the patient’s bedside. The first page contains instructions. These diaries are intended to create a clear narrative of non-clinical events that occurred during the patient’s stay in the intensive care unit (ICU). They are written in by family members, healthcare personnel, and, when possible, by the patients themselves, with the goal of helping ICU survivors fill in gaps in their memory.ICUM [63]: A 14-item self-administered test. The items investigate the quantity and type of memories before, during, and after the intensive care period. A checklist with definitions of the different types of memories of the ICU stay serves as an aid, allowing the patient to mark what they remember. Validated and used in Italy [64], the data from this tool are then compared with the data from the medical record.Mini-mental state examination (MMSE) [65]: A neuropsychological test used to assess intellectual efficiency and cognitive impairment. It is the most common screening tool for evaluating global cognitive functioning, although the diagnosis of cognitive impairment cannot be made with this tool alone. The test consists of 30 items that pertain to seven different cognitive areas: orientation in time, orientation in space, word registration, attention and calculation, recall, language, and constructive praxis. The total score ranges from 0 to 30 points. A score equal to or below 18 indicates severe cognitive impairment, a score between 18 and 24 indicates moderate to mild impairment, a score of 25 is considered borderline, and a score between 26 and 30 indicates normal cognitive function. The Italian version is commonly used [51], and recent updates have been made to its normative values [66].Hospital anxiety and depression scale (HADS) [67]: This self-reported questionnaire comprises 14 items and includes subscales for anxiety and depression. Each item is scored from 0 to 3, resulting in a total score ranging from 0 to 21 for each subscale. Patients with scores above 7 exhibit symptoms of anxiety and depression [68]. The scale has previously been employed in studies related to post-intensive care syndrome (PICS) [49] and is validated in Italian, with a Cronbach’s alpha HADS-A at 0.85 and HADS-D at 0.89 [69].SF-36 [70]: This generic health questionnaire assesses various domains of general health status and consists of 36 items. It is further divided into eight subscales: physical functioning, role limitations due to physical health problems, bodily pain, general health perception, vitality, social functioning, role limitations due to emotional problems, and mental health. The questionnaire yields two summary scores: the physical component summary (PCS), which encompasses physical functioning, role limitations due to physical health problems, bodily pain, and general health domains, and the mental component summary (MCS), including vitality, social functioning, role limitations due to emotional problems, and mental health domains. The SF-36 has been used to evaluate PICS [71] and has already been validated in Italian with Cronbach’s alpha coefficient consistently exceeding the recommended level of 0.70 (ranging from 0.77 to 0.93) [72].

Families

Socio-demographic questionnaire.Diary.Hospital anxiety and depression scale (HADS) [67].SF-36 [70].

Healthcare Professional

Socio-demographic questionnaire.Diary.

### 2.6. Sample Size

A total of 150 patients will be enrolled for the quantitative part of the study. The aforementioned number was calculated based on the total number of recoveries performed in the university hospital where the sampling will take place. Taking into account the inclusion and exclusion criteria, this totals 245 subjects, with a 95% confidence interval and a 5% significance level, considering an expected prevalence of 0.3%. For the qualitative section, purposive sampling will be employed, continuing until data saturation is achieved [73].

### 2.7. Analysis

Descriptive statistics, including mean, standard deviation, median, interquartile range, and frequencies, will be employed to summarize socio-demographic and clinical data. Multilevel analysis using linear regression will be conducted to identify predictors of quality of life (QOL). Data collected from dyads will be analyzed using the actor–partner interdependence model (APIM) to observe how one aspect influences another. Significance will be determined for all tests with a *p*-value (*p*) < 0.05.

For the analysis of diaries, Braun and Clarke’s thematic analysis approach [74] will be used in the qualitative design. Interpretive narrative research will be employed to analyze the diaries [59,60]. Thematic analysis will describe the data, select codes, and construct themes following the six steps outlined by Braun and Clarke [74]. The interpretive approach was deliberately chosen to analyze the data for their characteristics and to interpret social reality through the subjective viewpoints of the participants within the context where the reality is situated. The participant’s interviews will be audio-recorded and conducted using open-ended questions, as outlined in the methodology. These interviews will take place in locations chosen by the participants, providing them with the freedom to describe their experiences fully. Prior to conducting the interviews, researchers will engage in “bracketing” as part of “critical reflection” [62]. This process helps ensure rigor in data analysis.

During the interviews, researchers will also take note of environmental factors, participants’ non-verbal language, and their reflections. For the phenomenological analysis, sampling will continue until data saturation [73] is reached, indicating redundancy of experiences.

The interviews will be transcribed verbatim and integrated with field notes. Subsequently, researchers will conduct individual and meticulous analyses to extract main themes and any sub-themes. These extracted themes will then be compared among researchers and presented to participants for verification.

### 2.8. Ethical Considerations

The study has received approval from the Ethics Committee of the Central Area of the Calabria Region, Italy, with protocol registration number 65 on 16 March 2023. The study will be conducted in accordance with the principles of the Helsinki Declaration and will adhere to current regulations on clinical trials. Confidentiality and anonymity will be ensured by assigning an alphanumeric code to each participant. All sensitive data will be handled in accordance with the principles of Legislative Decree No. 196 of 30 June 2003, and in compliance with the General Data Protection Regulation (European Regulation No. 2016/679).

## 3. Expected Results

This study aims to generate data regarding the use and impact of diaries in an ICU among survivors to critical illness and among their family members. The intensive care unit diary is based on cognitive behavioral therapy principles, specifically the concept of trauma exposure [75]. It assists patients in processing traumatic events by exposing them to trauma memories [76]. By filling gaps in memory [77], the intensive care diary reduces the incidence of anxiety, depression [78], and post-traumatic stress disorder [19], thereby improving the quality of life of the survivors [48]. The importance of consistently assessing the perceived QoL in patients is a well-established concept in the healthcare literature. Longitudinal studies will enable us to identify predictors and interdependencies of this factor among patients and their relatives, contributing to the enhancement of nursing care for both patients and caregivers.

## 4. Implications for Practice

ICU survivors experience changes in their physical and mental health. The use of diaries in intensive care is associated with a lower incidence of psychological disorders, which are common after intensive care unit stay, and an improved perception of their own quality of life. This study will allow us to comprehend the impact of diaries on ICU survivors and their family members.

The utilization of diaries could lead to enhancements in the care provided, elevating its quality and fostering improved relationships among patients, family members, and healthcare professionals.

## Figures and Tables

**Table 1 healthcare-11-02583-t001:** Variables and instruments that will be used to collect the data: individual measures.

Operationalized as	Measured by	Patients	Parents	Healthcare Professional	Measured at (Months)
Socio-demographic	Socio-demographic questionnaire	X	X	X	0
ICUM	Memories	X			
Quality of life (over 18 years)	SF-36	X	X		0-3-12
Depression state	HADS	X	X		0-3-12
Anxiety	HADS	X	X		0-3-12
Cognitive state	MMSE	X			0-3-12

## Data Availability

Data sharing is not applicable to this article.

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
