# Peer review of "The Impact of an Intensive Care Diary on the Psychological Well-Being of Patients and Their Family Members: Longitudinal Study Protocol"

_healthcare, 2023, doi:10.3390/healthcare11182583_

Round 1

Reviewer 1 Report (Previous Reviewer 2)

Suggestions and comments:

-       The methodology section is not clear again. In the design reference is made to the mixed study but in the quantitative one it is not indicated that it is a descriptive observational through different types of questionnaires. Later, in the following sections it is only explained from the quantitative but not qualitative design. What techniques will be used (focal groups, interviews, etc)? What is the sample and qualitative sampling? Will the interviews be transcribed? etc

-       In the quantitative, how the sample has been calculated without indicating the target population that a descriptive conditions it.

Author Response

-Thank you  reviewer 1 for valuable suggestions. In this new version of  manuscript we have tried to follow all suggestions; we re certain that the manuscript acquires greater clarity and strength

In design section we've added: “Furthermore, in order to explore the experience and the meaning attributed to the diaries, Cohen's phenomenology  will also be used. This metodology combines  Husserl's descriptive and Gadamerian interpretative phenomenology. It is chosen as it has already been used given its ability to facilitate a deeper understanding of both lived experiences and the meanings attributed to these experiences” ;in analysis section we’ve added : “The partecipant's interviews  will be audio-recorded and will be conducted with an open question, just as foreseen by the methodology used, and conducted in a place chosen by the participants. The use of an open-ended question allows participants complete freedom to describe their experiences. As before the textual analysis of the diaries, even before conducting the interviews the researchers carry out the "bracketing".Such “critical reflection” and helps researchers to be rigorous when analyzing data.

During the interviews the researchers will note information about  environment,  non-verbal language of the participants and the reflections of the participants. For the phenomenological analysis, sampling will continue until data saturation:redundancy of experiences.The interviews, transcribed verbatim, will be integrated with field notes and will then be analyzed individually and meticulously by the researchers in order to extrapolate main themes and any sub-themes. The extrapolated themes will then be compared between the researchers and subsequently presented to the participants for verification.

-

Thank you for this valuable comment. We apologize for our lack. We have now specified what is required. In this new version of the manuscript you can read:” 150 patients will be enrolled for the quantitative part of the study. The above value was calculated considering total number of recovery performed in the University hospital where sampling will take place, which into account  inclusion and exclusion criteria, is quantified in 245 subjects, with a 95% confidence level and a 5% confidence interval, considering an expected prevalence of 0.3%.”

Reviewer 2 Report (New Reviewer)

Intensive care treatment is a serious challenge for the patient's family. The condition of the patient, specified in medical and psychological descriptions, edited after the end of treatment, does not always take into account the complexity of the experiences of the suffering person and his environment. Therefore, I consider the research protocol proposed in the article to be extremely important and necessary.

The theoretical assumptions of the research plan were very well formulated by the Authors.

However, I believe that the theoretical foundations of the planned research should be supplemented with a query of the literature on the subject from the years 2019-2022. During the Covid-19 pandemic, many articles have been published addressing the problem of treatment in intensive care units. I will not dare to suggest specific publications to the authors, but I definitely point to the need for an extended literature query.

The reviewed publication lacks clearly formulated research hypotheses. It is difficult for me to accept the study protocol without such an important element. This weakness of the research proposal should be corrected before publication.

It is also advisable due to the very essence or sense of publishing the study protocol. It is, among others, the possibility of replication and control of research by various scientific communities.

I highly appreciate the description of the planned research tools prepared by the authors, as well as the characteristics of the selection of the study groups.

Reading the study protocol, I got the impression that the authors attach great importance to the ethicality of the planned research. This deserves praise and recognition in the review. I would like to read a little more extensive comments by the authors on ethics in research conducted among former patients of intensive care units. This is an important aspect of the planned research. I leave the decision regarding the broadening of this issue to the Honorable Authors themselves.

In conclusion, I rate the presented research plan highly. The text was edited in an interesting and professional way. However, it requires some supplementation in terms of the content of the theoretical foundations of research and a clear formulation of the research problem, research questions and research hypotheses.

Author Response

Intensive care treatment is a serious challenge for the patient's family. The condition of the patient, specified in medical and psychological descriptions, edited after the end of treatment, does not always take into account the complexity of the experiences of the suffering person and his environment. Therefore, I consider the research protocol proposed in the article to be extremely important and necessary.

Thank you reviewer 2 for these statements which fully reflect the rationale for the study

The theoretical assumptions of the research plan were very well formulated by the Authors.

Thank you reviewer, we’re happy to read this

However, I believe that the theoretical foundations of the planned research should be supplemented with a query of the literature on the subject from the years 2019-2022. During the Covid-19 pandemic, many articles have been published addressing the problem of treatment in intensive care units. I will not dare to suggest specific publications to the authors, but I definitely point to the need for an extended literature query.

Thank you for your valuable feedback and suggestion regarding our planned research.

We have followed your suggestions and incorporated them following a literature review. We are open to receiving further recommendations to enhance the quality of our work.

The reviewed publication lacks clearly formulated research hypotheses. It is difficult for me to accept the study protocol without such an important element. This weakness of the research proposal should be corrected before publication. It is also advisable due to the very essence or sense of publishing the study protocol. It is, among others, the possibility of replication and control of research by various scientific communities.

Thank you reviewer for this important comment and we apologize for our unclearness. In this new version of  manuscript we have try to better explain what was requested.

In Study Objective we’ve added:” 1a) To test the hypothesis that an "intensive care diary" can reduce the patient's risk of developing symptoms of PTDS, anxiety-related disorders, depressive symptoms;

1b) To test the hypothesis that an "intensive care diary" can reduce the risk of devel-oping symptoms of PTDS, anxiety disorders, depressive symptoms in the relatives of pa-tients admitted to the ICU;

2) evaluate how the QoL trajectory of patients discharged from the ICU and their fam-ilies may be influenced by the use of an “intensive care diary”;

3) Explore the content and narrative structure of the diaries written by patients, rela-tives and health professionals;

4) explore the experiential experience regarding the use of a diary in intensive care.

In the hypothesis section we have now added: “Reconstructing memory of events that occurred during hospitalization in intensive care, through the use of a diary, can reduce onset of PTDS in subjects who survived to recovery ICU.

The use of a critical care diary, in which daily events can be recorded by family members and caregivers, can help reduce the occurrence of PTDS, anxiety, depressive symptoms and can promote psychological well-being in both patients than in their families.

Understanding how the use of diaries is experienced by partecipant  can help to develop educational programs aimed to implementing the use of this tool in order to improve the quality of life”.

I highly appreciate the description of the planned research tools prepared by the authors, as well as the characteristics of the selection of the study groups.

Thank you reviewer, we’re happy to read this comment

Reading the study protocol, I got the impression that the authors attach great importance to the ethicality of the planned research. This deserves praise and recognition in the review. I would like to read a little more extensive comments by the authors on ethics in research conducted among former patients of intensive care units. This is an important aspect of the planned research. I leave the decision regarding the broadening of this issue to the Honorable Authors themselves.

Thank you reviewer for this comment. We are glad you understood our attention and our work. In this version of manuscript we have described essential aspects, trying to be clear and to make people understand the attention paid to this aspect. The expansion of the question will be described in  presentation of results of, as planned by all the authors

In conclusion, I rate the presented research plan highly. The text was edited in an interesting and professional way. However, it requires some supplementation in terms of the content of the theoretical foundations of research and a clear formulation of the research problem, research questions and research hypotheses.

Thank you to  reviewer 2 for important suggestions. We are happy to read this positive evaluation. In this new version of  manuscript we have followed his suggestions, confident that this has given clarity and greater strength to the manuscript

Reviewer 3 Report (New Reviewer)

The article only presents a research design and not actual research. The topic is interesting, the review well-constructed, and the drawing itself well put together. But no research is presented, if the journal accepts protocols and research designs, you can accept the article for reviewing process.

The article is well written, however there are some typos to be corrected. 

Author Response

The article only presents a research design and not actual research. The topic is interesting, the review well-constructed, and the drawing itself well put together. But no research is presented, if the journal accepts protocols and research designs, you can accept the article for reviewing process.

We are happy to read that the reviewer 3 found the topic interesting. We pleased to read that  reviewer declares the entire manuscript is well constructed, and that he is satisfied with the review and design. We have submitted the manuscript to this eminent journal as it accepts research protocols.

Round 2

Reviewer 1 Report (Previous Reviewer 2)

-       The instruments in the manuscript that are collected from the patient are repeated on page 5 and 6. Delete one.

-       The rest of the issues have been clarified.

-       The instruments in the manuscript that are collected from the patient are repeated on page 5 and 6. Delete one.

-       The rest of the issues have been clarified.

Author Response

 The instruments in the manuscript that are collected from the patient are repeated on page 5 and 6. Delete one.

Done

We have submitted the manuscript to a native speaker translator for review in order to enhance the accuracy and clarity of the English.

Thank you, Reviewer 1, for dedicating your time to review our manuscript and for providing valuable suggestions. We have incorporated these suggestions into the revised version of our manuscript, aiming to enhance its clarity and comprehensiveness.

This manuscript is a resubmission of an earlier submission. The following is a list of the peer review reports and author responses from that submission.

Round 1

Reviewer 1 Report

Thank you for the opportunity to review this article.

While this is a subject of interest, below are some recommendations for tuning.

Abstract.

*The objective is different from the objectives presented in the body of the text.

*The keywords must be standardized writing. Ex. start with the first capital letter of each keyword.

Introduction

*Missing information that describes whether the use of diaries in the ICU is frequent.

Objective

*How will the impact of the diary be evaluated?

Design

*I found the study design a little confusing, especially in the qualitative part, this is not clear.

Exclusion criteria

*For health professionals: This information is redundant, as the inclusion criterion is to work in an ICU.

* HADS = The HADS scale evaluates SYMPTOMS of anxiety and depression, that is, patients with scores above 7 have SYMPTOMS of anxiety and depression

Sample size

*See article Sample Size in Clinical and Experimental Studies (Tamanho da amostra em estudos clínicos e experimentais).  By Hélio Amante Miot

References

*I suggest updating the references, some are quite old. Preferably for more recent articles

Reviewer 2 Report

Here are some comments and suggestions:

-       A study protocol should not include a discussion or conclusion as such unless a scoping review or systematic review is included. That must be modified in the document as relevance or implications or possible results and also in the abstract.

-       Keywords should be MeSH terms.

-       It is advisable to use full stops in the background, especially in the first paragraph.

-       The objective of the study can be unified in one including patient and family.

-       In the design section, first of all it should be indicated, as I understand it, that it is a mixed study. On the one hand, an analytical Cohorts observational study? and then the qualitative study that, like the quantitative one, must specify what design it is (phenomenological, ethnographic, ethnomethodological, etc.)

-       The professionals working in UCI could not speak and understand Italian?

-       Could you explain the exclusion criteria? It is not understood why drug users are excluded.

-       It is advisable to divide the instruments by family, patients and professionals. In addition, of the different scales used, it is due to carrying validation information on them, such as Cronbach's alpha.

-       In the simple size section it is stated that no sample calculation is necessary? In other words, in an analytical study we should not have a minimum number of participants per exposed group and non-exposed group? It is not necessary to indicate the type of sampling? Is it not necessary to indicate anything about the qualitative sample?

-       In data analysis it must be more extensive considering intervals of confidence and significances.

-       Although there should be no discussion, a section on expected results should be established where the literature related to similar studies and their fundamental conclusions are included. The section is now insufficient.

-       The conclusions should be implications for practice.